# Some Shape, Durability and Structural Strategies at the Conceptual Design Stage to Improve the Service Life of a Timber Bridge for Pedestrians

**Alessandra Fiore \*, Martino Antonio Liuzzi and Rita Greco**

Politecnico di Bari, DICAR, Via Orabona 4, 70125 Bari, Italy; martino.antonio.liuzzi@gmail.com (M.A.L.); rita.greco@poliba.it (R.G.)

\* Correspondence: alessandra.fiore@poliba.it



**Featured Application: The shape, durability and structural strategies proposed in this study for the design of timber bridges are aimed at helping professionals to better understand the issues connected to a proper service functioning of these structures, in order to realize constructions with good performance.**

**Abstract:** The use of wood in the construction of bridges has increased in recent decades thanks to the characteristics of this material, i.e., environmentally-friendly and suitability within natural landscapes. Nevertheless, timber constructions may be affected by degrading effects due to biological and/or abiotic agents, and may be exposed to impacts or vibrations due to external forces such as wind, earthquakes or walking pedestrians. Consequently, bridge performance with respect to these aspects should be assessed from the early design stage. Within this context, in this study, some shape, structural and durability strategies dealing with the design of timber bridges for pedestrians are investigated in order to extend the service life of these constructions. More precisely, a methodology consisting of three steps, to be applied at the early conceptual design stage, is proposed. The three fundamental steps to be considered in the preliminary design of timber bridges are: (i) main boundary constraints and load-bearing system; (ii) durability; (iii) vibration levels. In the study, the presented methodology is applied and described for the design of a pedestrian and cyclist timber bridge over the Gravina torrent, in Apulia (Italy).

**Keywords:** timber bridge; conceptual design; durability; modal calibration

---

## 1. Introduction

Until the sixteenth century, wooden bridges were mainly used for provisional constructions due to their vulnerability to atmospheric agents; the singular case of the Bassano del Grappa bridge in Palladio, which is still in use thanks to careful maintenance, is one such example.

From the second half of the nineteenth century onwards, materials such as iron, steel and reinforced concrete were preferred. Nevertheless, in recent decades, in many European countries, the use of wood in constructions, and specifically in bridges, was rediscovered thanks to the environmental sustainability of this material, the optimization of engineered wood such as glulam and related connections, and the development of better protection methods [1].

Italy is also experiencing this, in particular in the use of glulam bridges intended to be used by pedestrians and cyclists. The architectural choice of the uncovered bridge mainly characterizes these structures; direct exposure to atmospheric agents inevitably leads to the premature deterioration of structural components due to the biological nature of wood [2].

Designing bridges is a complex engineering task. In order to achieve a successful bridge design, engineers have to simultaneously optimize the safety, serviceability, durability, aesthetics and economy of the structure to be built. A good conceptual design is the result of knowledge, experience and intuition, combined with a clear idea of the flow of forces in structures, a preliminary estimation of loads and a first appraisal of erection techniques and equipment [3].

Thus, at the preliminary design stage, it is necessary to consider a number of factors such as the materials to be used, the geometry of obstacles, the foundation, the environmental and landscape conditions and the appearance of the bridge.

As outlined, the theory of conceptual design is very extensive; therefore, the proposed study focuses on some crucial aspects in the design process, specifically dealing with timber bridges, that may affect their service life, e.g., main constraints and load-bearing system, durability and vibration levels. So, the multiobjective design of a pedestrian and cycle timber bridge over the Gravina torrent in Apulia (Italy) is carried out by applying a new methodology based on the aforementioned items.

## 2. Conceptual Design

The conceptual design of a structure is a preliminary and critical step of design, during which essential features take form to optimize performance. The design process should be based on the fundamental principles of the required structural engineering, boundary conditions and functionality [3–5].

Within this framework, the proposed study deals with the preliminary conceptual design of a pedestrian and cycle timber bridge over the Gravina torrent in Apulia (Italy), even though the introduced design strategy is general and could be applied to the design of any timber bridge.

For this purpose, a new approach was developed and applied, consisting of including in the early stage of design an analysis of three crucial aspects that can significantly impact the performance and safety of timber bridges during their service life: (i) main boundary constraints and load-bearing system; (ii) durability; (iii) vibration levels.

The first fundamental step of the conceptual design of a timber bridge was identified in the appraisal of the boundary conditions (ground profile), and, in particular, of the span length to be covered, that can significantly affect the choice of the structural system, the structural height and the placement of the construction. By observing Figure 1 [6], it can be argued that, as the span grows, the structural configuration becomes progressively more complex, according to the sequence listed in the figure caption.

Regarding timber bridges, durability was identified as another critical issue to be considered in the conceptual design [7–10]. In [11], it is shown how the lack of a preliminary durability design can lead to early decay, i.e., after less than ten years of service life. Until now, in Italy, such a notion has been ignored by technicians, with consequent degradation of all existing wooden bridges.

According to Eurocode 5 [12], these structural components could be assigned to service class no. 3, that is, they could be assumed to be exposed to a moisture content > 20%, and therefore, to be vulnerable to fungi and insects; however, as proven by the conservation state of many existing bridges, this may not be sufficient to guarantee a good lifespan. The correct design approach, including durability devices, could be fundamental to improving the service life of these structures. For example, in other countries such as USA [13], directly exposed parts were avoided, meaning that covered bridges were generally favored [14].

Finally, an additional key aspect is the vibrational analysis of pedestrian timber bridges, which can experience severe serviceability issues under certain environmental loading scenarios [15,16]. These environmental actions are mainly associated with wind and earthquakes, as well as with walking pedestrians, which can result in strong impacts or vibrations [17–20].

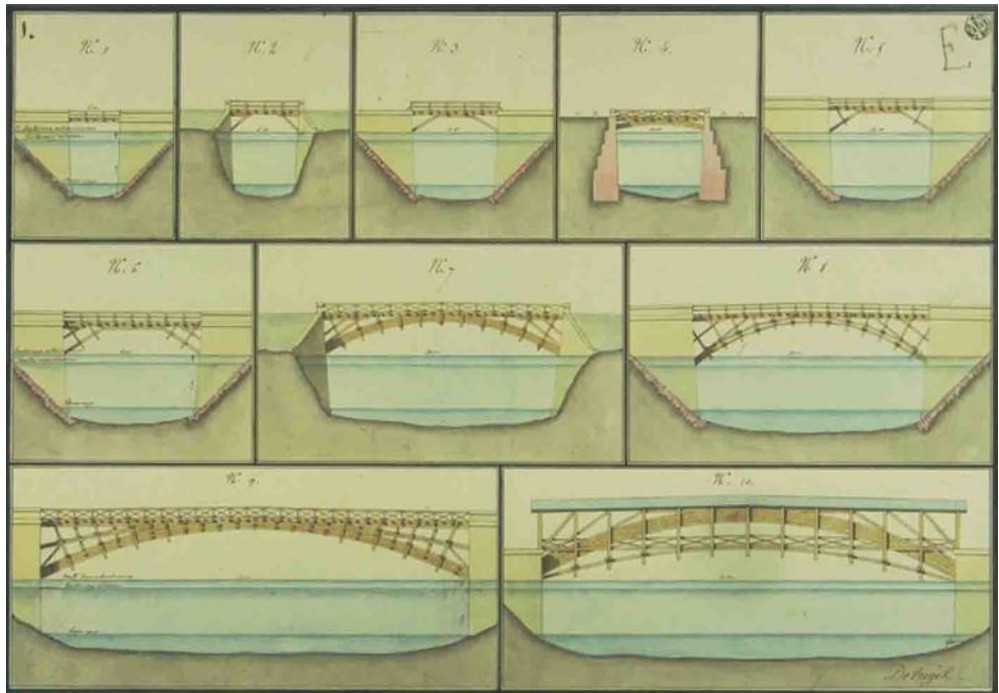

**Figure 1.** Influence of boundary conditions on the choice of a bridge structural system. "Concours de architecture presso l'École des Ponts et Chaussées", 1818. Student: Marie Fortunè de Vergés [6] (1. Simple girder; 2 and 3. girder supported by compressed struts; 4. girder reinforced by an additional. beam; 5 and 6 girder supported by compressed and branched struts; 7 and 8. girder supported by a multilayer arch connected through orthogonal wood elements; 9 and 10. More complex schemes).

A preliminary estimation of the expected vibration modes/levels would make it possible to avoid irregular deformations or dangerous coupling lateral-torsional oscillations [21], and therefore, could help to immediately classify pedestrian comfort level as acceptable or not [19,20].

These goals could be achieved using 3D finite element (FE) discretization, a tool that has been widely adopted within conceptual design from the early stages, with higher or lower levels of detail, making it possible to acquire the results of any design decision directly on a bridge 3D geometry.

The overall design procedure for timber bridges is summarized by the flow-chart in Figure 2. This is the first time that a design approach like this has been proposed in literature.

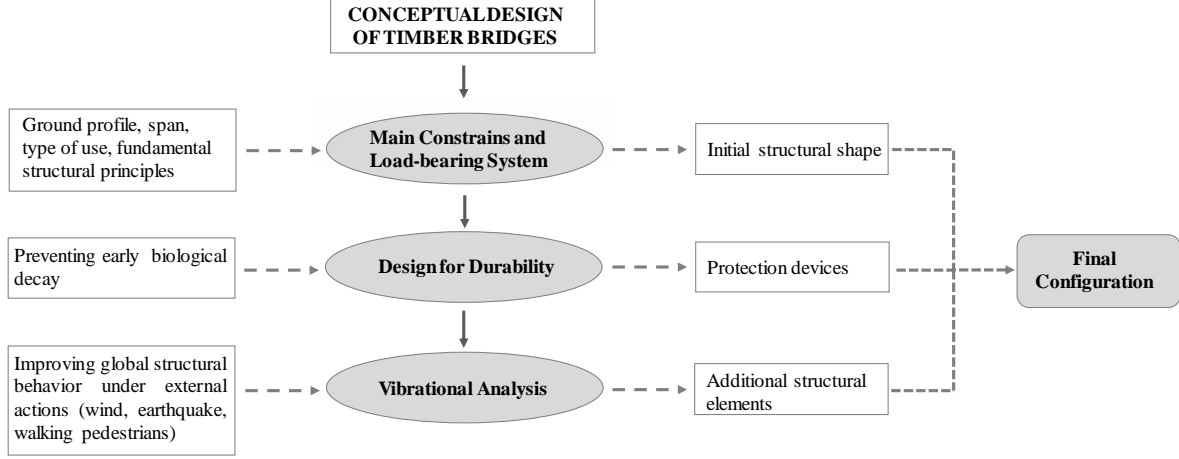

**Figure 2.** Flow-chart of the proposed methodology for the conceptual design of timber bridges.

### 3. Main Constraints and Load-Bearing System

The design process started by identifying the main boundary constraints.

The initial criterion for satisfying technical feasibility was the span, equal to 45 m; the position of supports was influenced by the topographic characteristics of the site and by the profiles of obstacles (Figure 3).

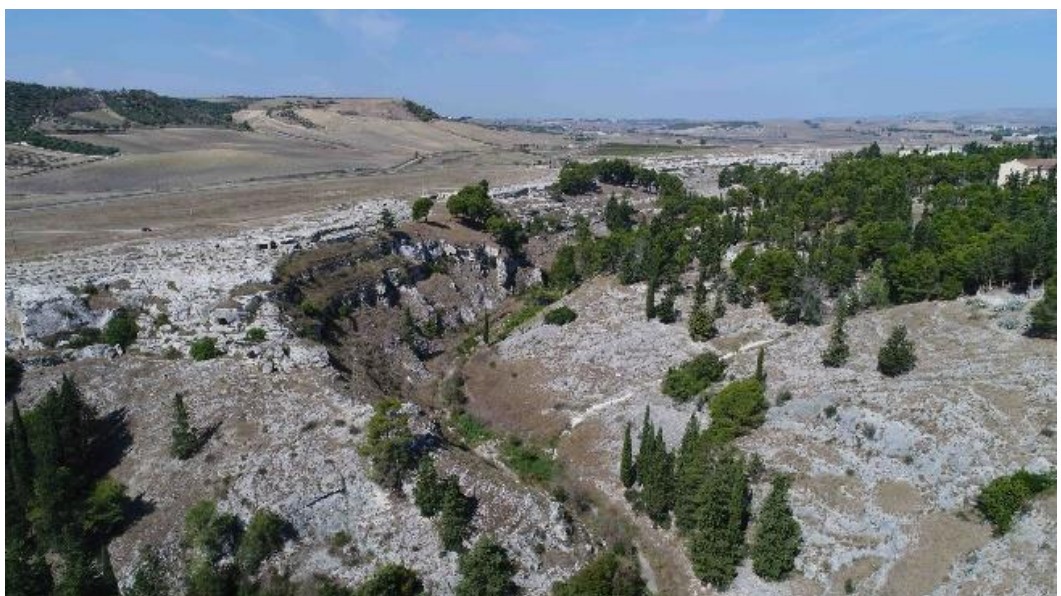

**Figure 3.** Topographic characteristics of the site in proximity of the Gravina torrent.

The first selected longitudinal profile was an arch lower-chord truss scheme (Figure 4a), which was chosen to provide suitable structural heights. It easily allowed the foundation to reach from the river level up to the road. Nevertheless, the resulting structure presented three main disadvantages: (i) it was very light, susceptible to oscillations (vertical and horizontal) and had little torsional rigidity [22]; (ii) maximum bending forces reached excessive values at midspan; (iii) the arch truss system was unprotected against weather conditions. Therefore, this proposal was abandoned.

The obvious solution was to adopt an arch lower-chord strut system with V-struts (Figure 4b). On the one hand, this second solution better addressed the environmental impact of the foundations and led to smaller bending moment effects, but on the other hand, it was not able to satisfactorily fulfill the robustness and durability requirements. In fact, the V-struts were excessively exposed to weather, especially to rainfall, with the consequent risk of a reduction of their bearing capacity.

For these reasons, a covered Pratt-type truss system was finally selected, by assuming the lower chord as cycle path level and the upper chord as the roof (Figure 5). Such a scheme fits well in the natural environment around the Gravina torrent, which is dominated by the concept of horizontality. In order to both increase the stiffness and improve durability, a secondary steel truss was placed on the main laminated-timber Pratt-type truss.

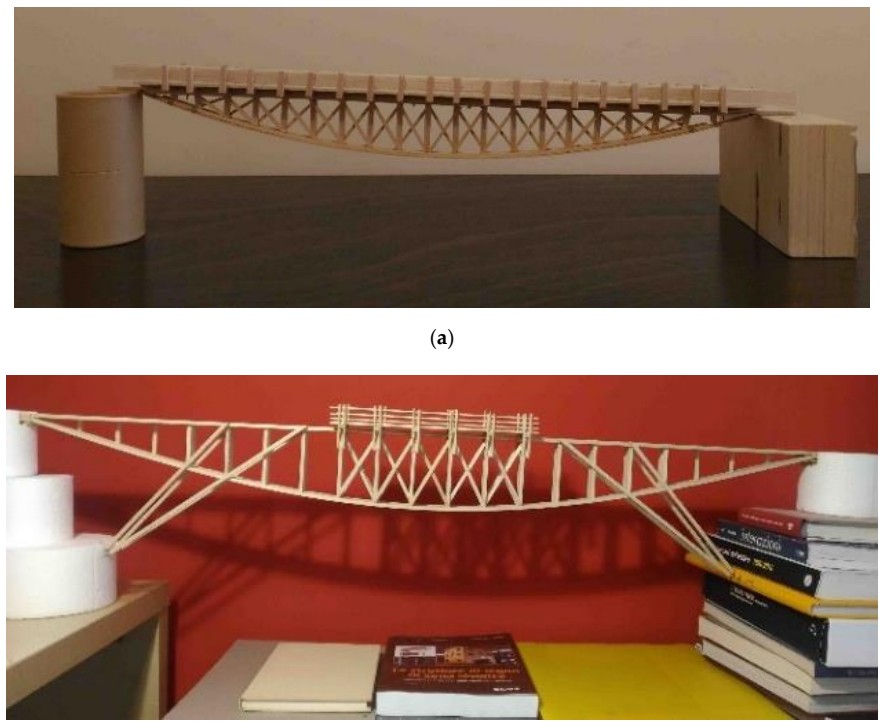

(**a**)

(**b**)

**Figure 4.** First longitudinal profiles for the timber bridge: (**a**) arch lower-chord truss scheme; (**b**) arch lower-chord strut system with V-struts.

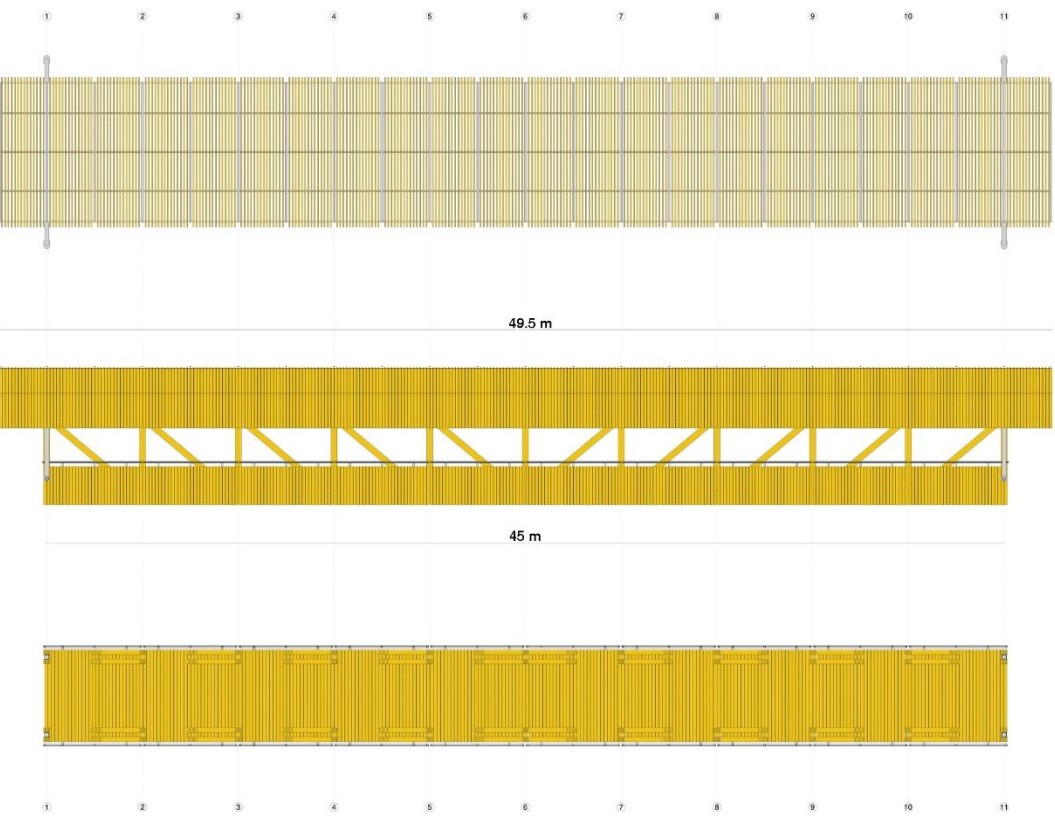

**Figure 5.** Final longitudinal profile chosen for the bridge.

As for the installation aspects, the bridge is located in an area where common construction site vehicles cannot be used, i.e., crane trucks and/or telehandlers; the use of a specifically designed crane would risk being as expensive as the entire work. For these reasons, assembly via helicopter would be appropriate.

The structure should therefore be divided into parts (to be assembled on site) with a weight equal to the maximum capacity of the chosen vehicle (for example the maximum capacity of the AS 332 Super Puma mode, a helicopter for aerial work, is 45 kN).

## 4. Design for Durability

According to Eurocode 0 [23], "The durability of a structure is its ability to remain suitable for use during its useful life, provided it is adequately maintained". The UNI EN 1001-2 [24] standard defines the durability of wood and wood-based products as the "resistance of the material to degradation induced by wood-borne organisms", including molds and fungi that cause its degradation by attacking the material under certain humidity conditions.

Eurocode 5 [12] defines three important service classes in order to assign the resistance and deformation values of the structural components, and therefore, of timber constructions in general.

With the aim of evaluating the moisture content, which is particularly important for biological durability, the European Standard EN 335 [25] also defines five use classes, i.e., five environmental conditions in which the wood can be found. The concept of use class is related to the probability that a wooden element is attacked by biological agents, and mainly depends on its moisture content, that should not exceed 20%.

The two classification systems differ in their consideration of the effects of humidity on the wood, i.e., of the mechanical type and the biological type, respectively. It should be noted that the classes of service and use are not performance classes; therefore, no indications are given on how long the wooden structures will remain in service. Similarly, the codes do not furnish any precise method by which to assess durability. In addition to the aforementioned codes, the European Standard EN 350-2 [26] only provides information about the natural durability of the wood species existing in Europe, while the European Standard EN 460: 1995 [27] indicates the requirements for wood corresponding to each use class.

Within this framework, in this study, the following approach is suggested to properly account for durability, and so, improve the bridge service life: (1) definition of the design durability; (2) individuation of the use class according to [25,26]; (3) choice of a design strategy, including protection measures and maintenance plan; (4) verification of durability on the basis of the rules proposed in [28]. The latter can be summarized in a set of design principles called the 4Ds: deflection, drainage, drying and durable materials [28]. Deflection concerns rain penetration control and includes all devices that minimize rainwater loads on the construction envelope, while drainage, drying and durable materials are concepts dealing with the management of water after it has reached the envelope.

The above strategy was applied to the pedestrian bridge described in this study. Durability was chosen based on the service life of the bridge, assumed equal to 50 years. Successively, focusing on the environmental context where the construction is planned, the third use class was set, corresponding to a moisture content > 20%, and so, to wood elements which are vulnerable to the attacks from fungi and insects.

The consequential choice of the service class depended not so much on the design environmental conditions, i.e., the structural components in laminated wood which are protected by the roof, but from the predictable conditions that could involve structural elements if the "water barrier", created by the lining vertical wooden staves, were lost. Furthermore, the Gravina stream, which flows at about 10 m below the bridge, although often dry, could lead to a moisture content > 85%. For these reasons, it was decided to design the bridge according to service class 3. On the basis of the above observations, the larch was chosen as the wood species to be used for the bridge structure.

Focusing on the design strategy for wood protection, it was fundamental to work on the cross section of the bridge, trying to apply the so-called 4D rule, known in the design of wooden buildings in the American context, but more recently also applied in Europe [28]. The spatial reticular steel truss, being cantilevered, protects the timber truss from atmospheric agents. As shown in Figure 6, the inclination of rain, assumed at an angle of 60° with respect to the vertical axis, is the design parameter according to which the cross section of the bridge and its construction details were modeled.

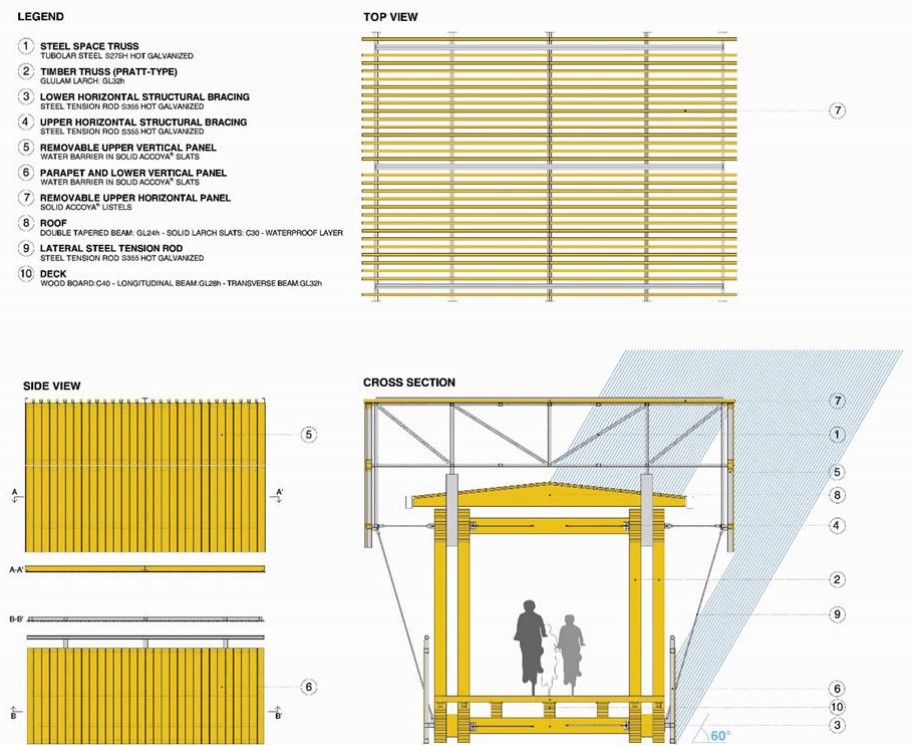

**Figure 6.** Design for durability of the timber bridge.

The hot-galvanized steel truss was calculated to withstand maintenance loads, avoiding the need to mount a scaffold to inspect the bridge; as a consequence, the side vertical panels were designed, in turn, to be easily removed from the top.

Being constituted by juxtaposed and overlapping staves, vertical panels absolve the dual role of constituting a "water barrier" and of creating shading, so as to avoid the phenomenon of delamination (Figure 6). The wooden vertical panels were forecasted in Accoya®, a molecularly-modified acetylated wood, characterized by a very low capacity to absorb water. Accoya® is associated with the highest durability class (1), corresponding to a service life of 50 years.

According to the above design strategy for durability, on the basis of the rules proposed in [28], the bridge should not be directly exposed to rainfall under normal service conditions, and thus, the moisture content and temperature should not reach values whereby wood decay occurs [2,11].

As for the protection of wood elements at the ends of the bridge, it is assumed that they are at a distance equal to 30 cm from the ground, thanks to suitably interposed supports.

In addition, it is supposed that occasionally the bridge could experience conditions of air humidity greater than 90% and temperatures higher than 20 °C, especially during summer storms. The construction details of the bridge cannot prevent moisture absorption under such circumstances. Nevertheless, considering that the aforementioned conditions do not occur frequently, and that the ventilation of structural elements and nodes is guaranteed, it can be assumed that the absorbed humidity could be easily released naturally.

## 5. Structural Behavior and Vibrational Analysis

Structural verifications were carried out according to the code prescriptions provided by the Italian technical code [29], Eurocode 5 [12] and CNR-DT 207 R1/2018 [30].

The materials used in the project are: solid larch C30, C40 [31] and glulam larch GL24h, GL28h, GL32h [32] for wood; and (ii) S275H and S355 [29] for steel. The choice of the wood species significantly affects the structural performance of each element. The bridge belongs to the second category; therefore, the considered traffic actions are: a concentrate load equal to 10 kN with a 0.10 m side square-footprint used for local verifications (load diagram 4); a crowd load, including dynamic effects equal to 5 kN/m$^2$ (load pattern 5) (Figure 7); a roof maintenance overload equal to 0.5 kN/m$^2$; and a crowd thrust pushing on the handrail equal to 1.5 kN/m.

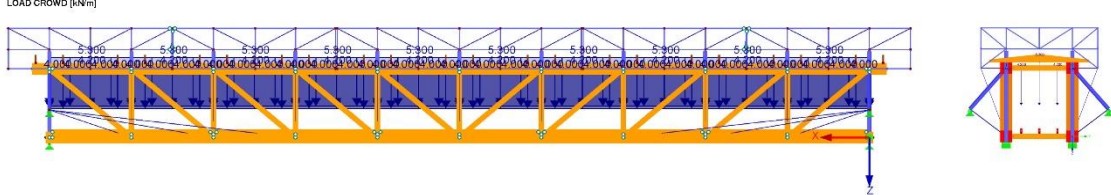

**Figure 7.** Representation of the crowd load.

The structural permanent loads of all elements were determined using FE calculation software. The specific weight of the chestnut wooden floor constituting the planking level is 4.84 kN/m$^3$, while that of the Accoya® covering is equal to 5.1 kN/m$^3$.

As shown in Figure 8, the main load-bearing system is constituted by two laminated-timber Pratt-type truss longitudinal beams that support the transverse frame (deck), the wooden roof and the steel spatial truss system. The steel space system includes tubular trusses in both longitudinal and transversal directions, with square-hollow or circular cross-sections in order to suitably overcome constructional issues. The values of the main mechanical properties of the materials, used for numerical simulation, are reported in Table 1.

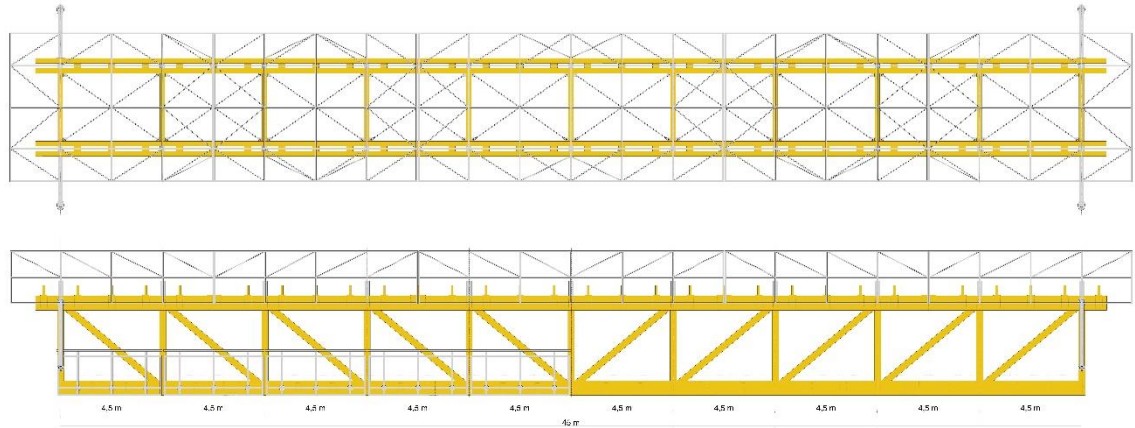

**Figure 8.** Structural system of the timber bridge over the Gravina torrent.

A three-dimensional FE model of the bridge was built by using RSTAB in the Dlubal software. Only structural elements were included in the model; nonstructural elements were considered as extra masses. The model has a total of 928 nodes and 2721 frame elements. The modeling of the glulam truss was carried out by schematizing the joints as rigid nodes.

**Table 1.** Mechanical properties of materials (GL24h, GL28h, GL32h: Glued Laminated Timber grades; $f_{t,0,k}$: characteristic tension strength parallel the grain; $f_{c,0,k}$: characteristic compression strength parallel the grain; S275H: Steel Grade according to UNI EN 10210-1; S355: Steel Grade according to UNI EN 10025-2; $f_{yk}$: characteristic yield strength; $f_{tk}$: characteristic failure strength; $E$: Young's modulus; $\gamma$: specific weight).

|  | GL24h | GL28h | GL32h |  | S275H | S355 |
|---|---|---|---|---|---|---|
| $f_{t,0,k}$ [kN/cm$^2$] | 1.92 | 2.23 | 2.56 | $f_{yk}$ [kN/cm$^2$] | 27.5 | 35.5 |
| $f_{c,0,k}$ [kN/cm$^2$] | 2.4 | 2.8 | 3.2 | $f_{tk}$ [kN/cm$^2$] | 43 | 51 |
| $E$ [kN/cm$^2$] | 1150 | 1260 | 1420 | $E$ [kN/cm$^2$] | 21000 | 21000 |
| $\gamma$ [kN/m$^3$] | 4.2 | 4.6 | 4.9 | $\gamma$ [kN/m$^3$] | 78.5 | 78.5 |

It is worth noting that despite the uncertainties which are intrinsic in a FE model, the performed analysis, described in the following section, satisfactorily explain the main features of the proposed methodology.

*Modal Calibration*

The results of the modal analysis are summarized in Table 2, while the initial bridge configuration is reported in Figure 9. In particular, Table 2 shows the fundamental modes of vibration in the transversal ($y$) direction ($T = 0.335$ s), in the vertical ($z$) direction ($T = 0.218$ s) and in the torsional direction around the longitudinal axis ($x$) ($T = 0.239$ s; $m@jX$=191285.51 kg·m$^2$). The torsional mode of vibration around the longitudinal axis is depicted in Figure 10a; it can be observed that the corresponding cross-section deformation is nonhomogeneous and assumes a rhomboidal shape.

**Table 2.** Results of the modal analysis of the original model ($m_{eX}$, $m_{eY}$, $m_{eZ}$: translational masses; $m_{@jX}$, $m_{@jY}$, $m_{@jZ}$: rotational masses).

| | Modal Mass | Effective Modal Mass | | | | | | Modal Participating Mass Ratio | | | Nat. Freq. | Nat. Per. |
|---|---|---|---|---|---|---|---|---|---|---|---|---|
| n$_r$. | $M_i$ [kg] | $m_{eX}$ [kg] | $m_{eY}$ [kg] | $m_{eZ}$ [kg] | $m_{@jX}$ [kgm$^2$] | $m_{@jY}$ [kgm$^2$] | $m_{@jZ}$ [kgm$^2$] | $\rho_{meX}$ | $\rho_{meY}$ | $\rho_{meZ}$ | $f$ [Hz] | $T$ [s] |
| 1 | 21094.16 | 0.00 | 50308.9 | 0.00 | 4158.6 | 0.00 | 2.28 | 0.000 | 0.662 | 0.000 | 2.989 | 0.335 |
| 2 | 15913.94 | 0.00 | 6975 | 0.00 | 191285 | 0.00 | 120.83 | 0.000 | 0.092 | 0.000 | 4.178 | 0.239 |
| 3 | 34247.83 | 0.00 | 0.00 | 54767.6 | 0.77 | 118.43 | 0.00 | 0.000 | 0.000 | 0.721 | 4.580 | 0.218 |
| 14 | 12602.66 | 52917.5 | 0.00 | 0.00 | 0.01 | 1182525 | 0.97 | 0.696 | 0.000 | 0.000 | 15.463 | 0.065 |

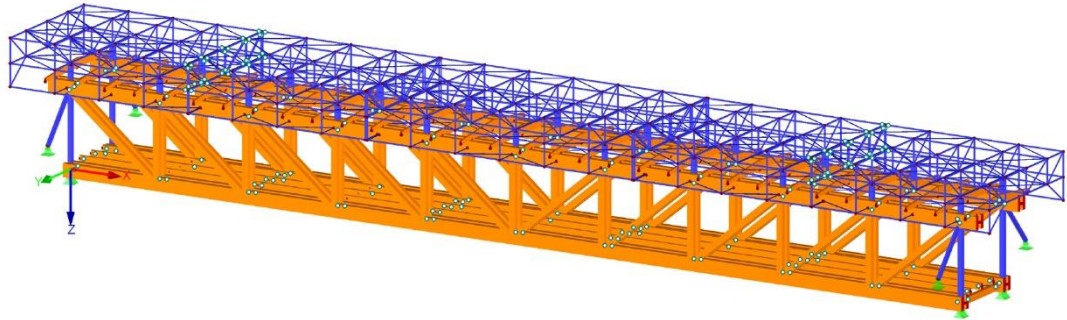

**Figure 9.** Three-dimensional model of the initial configuration.

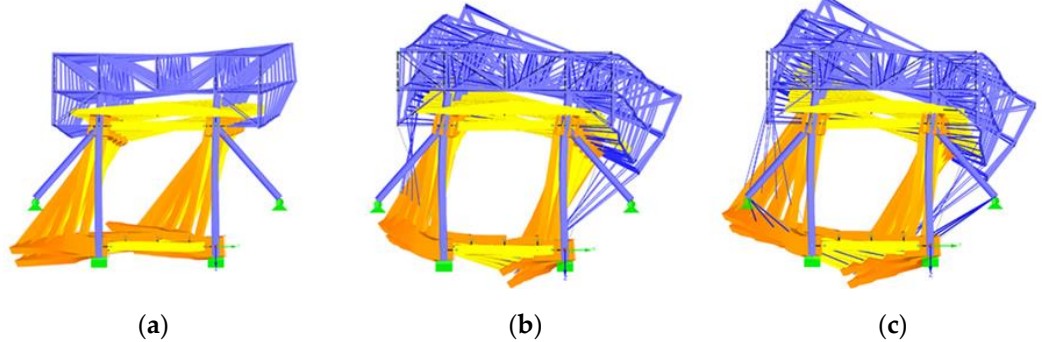

(**a**)　　　　　　　　　　　(**b**)　　　　　　　　　　　(**c**)

**Figure 10.** Cross-section of the torsional modal shape: (**a**) original model (2nd mode of vibration, $T$=0.239 s; $m_{@jX}$ = 191285.51 kg·m$^2$); (**b**) after the first model update (3rd mode of vibration, $T$ = 0.197 s; $m_{@jX}$ = 466015.09 kg·m$^2$); (**c**) after the second model update (3rd mode of vibration, $T$ = 0.17 s; $m_{@jX}$ = 421632.16 kg·m$^2$).

In order to avoid such a deformed configuration, lateral steel tension rods were added on both sides of the bridge along the entire span, thereby conferring stability to the horizontal section (Figures 10b and 11). The results of the modal analysis of the new model are reported in Table 3; Figure 10b shows the cross-section deformation of the torsional mode of vibration ($T$ = 0.197 s; $m@jX$ = 466015.09 kg·m$^2$), which was homogeneous after the above adjustment. Moreover, it can be argued that the modal participating mass ratios of the fundamental modes of vibration in the three directions are significant (~70%), indicating that there is no excessive dispersion of mass among oscillation modes, and that the structure is well organized [33,34].

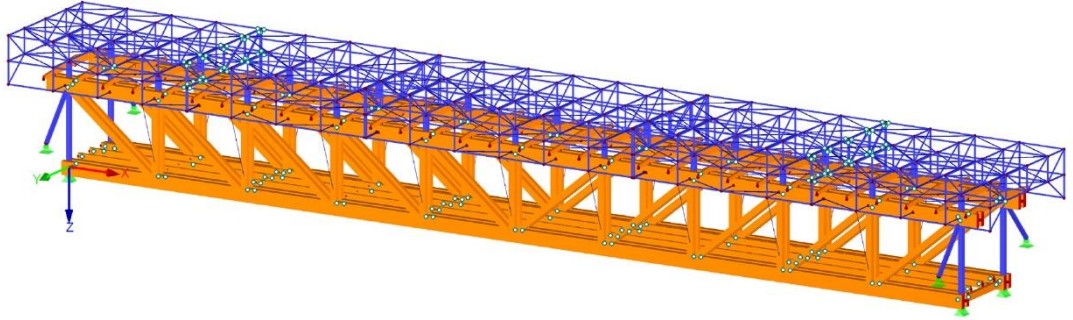

**Figure 11.** Three-dimensional model after the first update.

**Table 3.** Results of the modal analysis after the first model update ($m_{eX}$, $m_{eY}$, $m_{eZ}$: translational masses; $m_{@jX}$, $m_{@jY}$, $m_{@jZ}$: rotational masses).

| n$_r$. | Modal Mass $M_i$ [kg] | Effective Modal Mass $m_{eX}$ [kg] | $m_{eY}$ [kg] | $m_{eZ}$ [kg] | $m_{@jX}$ [kgm$^2$] | $m_{@jY}$ [kgm$^2$] | $m_{@jZ}$ [kgm$^2$] | Modal Participating Mass Ratio $\rho_{meX}$ | $\rho_{meY}$ | $\rho_{meZ}$ | Nat. Freq. $f$ [Hz] | Nat. Per. $T$ [s] |
|---|---|---|---|---|---|---|---|---|---|---|---|---|
| 1 | 32600.03 | 0.00 | 58121.09 | 0.00 | 679.33 | 0.01 | 0.21 | 0.000 | 0.763 | 0.000 | 3.169 | 0.316 |
| 2 | 34323.16 | 0.00 | 0.00 | 54911.9 | 0.78 | 118.15 | 0.00 | 0.000 | 0.000 | 0.721 | 4.576 | 0.219 |
| 3 | 13592.08 | 0.00 | 6.53 | 0.00 | 466015 | 0.00 | 903.1 | 0.000 | 0.000 | 0.000 | 5.083 | 0.197 |
| 14 | 12619.2 | 52992.5 | 0.00 | 0.00 | 0.01 | 1188580 | 1.04 | 0.696 | 0.000 | 0.000 | 15.459 | 0.065 |

From Table 3, it can be noted that the value of the natural period of the fundamental vertical mode of vibration ($T$ = 0.219 s) is near that of the torsional mode of vibration ($T$ = 0.197 s), and consequently, if vertical mode occurs, the torsional one could also simultaneously activate, causing dangerous vibrations in the structure. So, it is suggested to introduce into the model additional steel cables, represented in Figures 10c and 12, in order to make the translational and the torsional modes of vibration asynchronous.

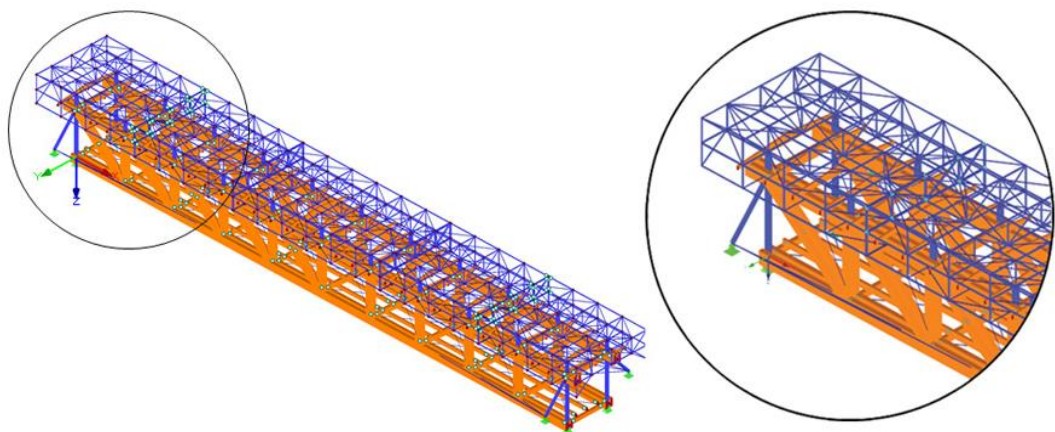

**Figure 12.** Three-dimensional model after the second update, with enlargement of the inserted cables.

The results of the modal analysis after this second model update are reported in Table 4. After the second modal correction, the percentage gap in terms of periods between vertical and torsional modes of vibration increases from 8.78% to 21% with respect to the initial model and from 10% to 21% with respect to the first modal update.

**Table 4.** Results of the modal analysis after the second model update ($m_{eX}$, $m_{eY}$, $m_{eZ}$: translational masses; $m_{@jX}$, $m_{@jY}$, $m_{@jZ}$: rotational masses).

| | Modal Mass | | Effective Modal Mass | | | | | | Modal Participating Mass Ratio | | | Nat. Freq. | Nat. Per. |
|---|---|---|---|---|---|---|---|---|---|---|---|---|---|
| $n_r$. | $M_i$ [kg] | $m_{eX}$ [kg] | $m_{eY}$ [kg] | $m_{eZ}$ [kg] | $m_{@jX}$ [kgm²] | $m_{@jY}$ [kgm²] | $m_{@jZ}$ [kgm²] | $\rho_{meX}$ | $\rho_{meY}$ | $\rho_{meZ}$ | $f$ [Hz] | $T$ [s] |
| 1 | 19268.17 | 0.00 | 51966.2 | 0.00 | 32594 | 0.00 | 84.55 | 0.000 | 0.675 | 0.000 | 3.721 | 0.269 |
| 2 | 34445.37 | 0.00 | 0.00 | 55375.5 | 0.77 | 116.31 | 0.00 | 0.000 | 0.000 | 0.720 | 4.635 | 0.216 |
| 3 | 18015.99 | 0.00 | 4181.06 | 0.00 | 421632 | 0.00 | 1309 | 0.000 | 0.054 | 0.000 | 5.890 | 0.170 |
| 14 | 9177.12 | 59315.2 | 0.00 | 0.00 | 0.01 | 376924.1 | 0.54 | 0.773 | 0.000 | 0.000 | 16.501 | 0.061 |

The adoption of the above modifications could safeguard the bridge from irregular deformations and from dangerous coupling lateral-torsional vibration modes under external loads such as wind or earthquakes [17,18,21], thereby enhancing its service performance.

Finally, it can be verified that the final structural configuration of the bridge fulfills the limits imposed by technical codes with regard to the vibrations due to walking pedestrians. In fact, common physical activities such as jumping, running and walking on footbridges may produce dynamic effects, and thus, significant vibrations [19,20].

The assessment of the compliance of the main required comfort constraints in the vertical and horizontal directions, according to Eurocode 0 [23] and Eurocode 5 [12], is summarized in Table 5.

The comfort criteria are defined in terms of the maximum acceptable acceleration of any part of the deck. The maximum acceptable values of acceleration are 0.7 m/s² for vertical vibrations and 0.2 m/s² for horizontal vibrations. Such verification of the comfort criteria should be performed only if the deck fundamental frequency is less than 5 Hz for vertical vibrations and less than 2.5 Hz for horizontal vibrations.

In conclusion, assessing the acceptability of these vibrations from the early conceptual design stage could help to avoid both discomfort to pedestrians and deterioration of the structural integrity during the bridge's service life.

**Table 5.** Verification of comfort criteria (*T*: period; *f*: frequency; *M*: total mass of the bridge; *ξ*: damping ratio).

| Vertical Vibration | | | $a_{vert} \leq 0,7$ m/s$^2$ |
|---|---|---|---|
| **2nd mode** | | | |
| *T* [s] | $f_{vert}$ [Hz] | *M* [kg] | $\zeta$ |
| 0.216 | 4.635 | 94531 | 0.015 |
| walking pedestrian | | if $f_{vert} < 5,0$ Hz | |
| $a_{vert,1}$ [m/s$^2$] = | $100/M\zeta$ = | 0.07 | VERIFIED |
| **Horizontal Vibration** | | | $a_{hor} \leq 0,2$ m/s$^2$ |
| 1st mode | | | |
| *T* [s] | $f_{hor}$ [Hz] | | |
| 0.269 | 3.721 | | |
| walking pedestrian | | $f_{hor} > 2,5$ Hz VERIFIED | |

## 6. Conclusions

In this study, a new methodology to carry out the conceptual design of timber bridges was proposed and described. A possible pedestrian and cycle bridge over the Gravina river, in Apulia (Italy), was considered as a case study.

The proposed approach consists of assessing, from the early stage of design, the performance of timber bridges concerning three fundamental aspects: (i) the main boundary constraints and load-bearing system; (ii) durability; (iii) vibration levels. Such a strategy is novel in the scientific and technical literature.

The structural scheme of the bridge over the Gravina river, the materials and the covering systems were chosen taking into account multiple aspects, none of which is negligible, by firstly analyzing the boundary conditions and structural behavior, and subsequently, durability issues.

Finally, a vibrational analysis of the bridge was carried out by a two-step model update. More precisely, the finite element model of the bridge was corrected on the basis of the modal analysis, by introducing additional steel cables in order to reach two main objectives: (i) a homogeneous cross-section modal shape corresponding to the torsional mode of vibration; (ii) making translational and torsional modes of vibration asynchronous.

The suggested shape, durability and structural elements were chosen to maximize the long-term behavior of the bridge under external loads, thereby avoiding excessive stresses, deterioration and dangerous coupling lateral-torsional deformation.

**Author Contributions:** A.F. conceived the initial idea of introducing a new methodology for the conceptual design of timber bridges and in particular developed the vibrational analysis and the strategy of structural modal calibration. M.A.L. developed the durability strategy and implemented the models of the bridge in all steps (identification of main load-bearing system; durability model; Finite Element models). R.G. gave a contribution in the interpretation of the results. All authors have read and agreed to the published version of the manuscript.

**Funding:** This research received no external funding.

**Conflicts of Interest:** The authors declare no conflicts of interest.

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
