# Peer review of "Some Shape, Durability and Structural Strategies at the Conceptual Design Stage to Improve the Service Life of a Timber Bridge for Pedestrians"

_applsci, doi:10.3390/app10062023_

Round 1

Reviewer 1 Report

The paper presents a structural solution to improve the durability of a timber pedestrian bridge located in Apulia, Italy. The strength of the work is that the authors made quite well justified and thorough study using a set of appropriate preparation and characterization techniques, enabling to get convincing novel data in this subject. The weakness is that the work lacks some information as:

The introduction must be improved because do not provide sufficient background. 2, paragraphs 56-58 – add more details about structural systems from Fig. 2 (explain the types of variants) 2, regarding to Fig. 3 a and b, what are the scientific arguments that the two proposed structures do not resist? Between the two variants of Fig. 3 and the final version there is a very big difference from a constructive point of view. Improve Fig 4 as quality of image and with explanation about main parts of bridge (lower chords, upper chords, secondary steel truss) and the overall dimensions. The paragraphs 97-125 (p. 4) are more appropriate in the introductory chapter. In the chapter. 3, is not mentioned nothing about the protection of wood elements of the ends of bridge which are in contact with soil. In fig. 5, the quality of cross section with numbers must be improved. In the chapter. 4, the description of loading can be exemplified by graphical representation. Each figure from Fig 6 must be explained. In all titles of tables, the abbreviations (symbols) should be explained. In subsection 4.1. you discussed about two updating of bridge. Can you provide some images with all three variants (base model and the two updating), without deformations? The interpretation of the results is not satisfactory. How much did the structure improve after the second upgrade over the initial proposal (percentage)? Based on the Eurocode 5 calculations, do the simulation results fall within the allowable limits? The references must be extended.

Author Response

Manuscript ID: applsci-719020,

Title: SOME SHAPE, DURABILITY AND STRUCTURAL STRATEGIES AT THE CONCEPTUAL DESIGN STAGE TO IMPROVE THE SERVICE LIFE OF A TIMBER BRIDGE FOR PEDESTRIANS

Applied Sciences

Special Issue: Assessing and Extending the Service Life of Bridges

ANSWER TO REVIEWERS' COMMENTS

All the new/changed parts in the manuscript are outlined in red. In the following the comments of Reviewer 1 are answered. The authors thanks Reviewer 1 for his comments that surely helped to improve the quality of the paper.

Referee: 1

Comments to the Author

The paper presents a structural solution to improve the durability of a timber pedestrian bridge located in Apulia, Italy. The strength of the work is that the authors made quite well justified and thorough study using a set of appropriate preparation and characterization techniques, enabling to get convincing novel data in this subject. The weakness is that the work lacks some information as:

The introduction must be improved because do not provide sufficient background. The paragraphs 97-125 (p. 4) are more appropriate in the introductory chapter. The references must be extended.

Answer: The introduction has been entirely reformulated and Section 2 by the title “Conceptual design” has been introduced in order to better explain the methodology proposed in the paper. References have been extended.

2, paragraphs 56-58 – add more details about structural systems from Fig. 2 (explain the types of variants) 2, regarding to Fig. 3 a and b, what are the scientific arguments that the two proposed structures do not resist? Between the two variants of Fig. 3 and the final version there is a very big difference from a constructive point of view. Improve Fig 4 as quality of image and with explanation about main parts of bridge (lower chords, upper chords, secondary steel truss) and the overall dimensions..

Answer: Structural system in Fig 2 (in the revised version Fig. 1) has been explained in detail. More arguments have been used to explain the motivations that led to the third and final scheme. The quality of Fig. 4 (in the revised version Fig. 5) has been improved. The main parts of the bridge are described in Fig. 6.

In the chapter. 3, is not mentioned nothing about the protection of wood elements of the ends of bridge which are in contact with soil. In fig. 5, the quality of cross section with numbers must be improved.

Answer: It has been explained that: “As to the protection of wood elements at the ends of the bridge, it is assumed that they are at a distance equal to 30 cm from the ground, thanks to suitable interposed supports.” The quality of Fig. 5 (in the revised version Fig. 6) has been improved.

  In the chapter. 4, the description of loading can be exemplified by graphical representation. Each figure from Fig 6 must be explained. In all titles of tables, the abbreviations (symbols) should be explained. In subsection 4.1. you discussed about two updating of bridge. Can you provide some images with all three variants (base model and the two updating), without deformations? The interpretation of the results is not satisfactory. How much did the structure improve after the second upgrade over the initial proposal (percentage)? Based on the Eurocode 5 calculations, do the simulation results fall within the allowable limits?

Answer: As requested, a graphical representation of loading has been introduced in Fig. 7. Each figure from Fig 6 (in the revised version Fig. 8) is explained. The images of all three variants are provided (Figs. 9, 11, 12). The interpretation of the results has been improved. The percentage improvements after the second upgrade have been inserted in the analysis. Also verifications according to Eurocode 5 have been included (Table 5).

Reviewer 2 Report

1 English communication is sufficient for comprehension of the authors' intent. => Needs professional proofreading.

2 Literature review is sparse, and tends to be factual rather than critiquing the body of knowledge. =>  Literature needs work.

3 The work is presented as a design report. The methodology is implied rather than explicit. =>  Add a method statement.

4 A number of design solutions were initially considered, and one selected. However we are not informed adequately about how this decision was made. Figures are provided for some wooden models, though these are not particularly informative except in a descriptive way. The overall effect is that the initial stages of concept design appear to have been ad-hoc. => Manuscript needs a better and more rational explanation of the decision-making.

5 A detailed analysis of the selected bridge design is presented. For the most part this appears to be design-as-usual. => If there is any novel academic contribution here, it needs to be more explicit.

6 There is no critical review of the assumptions or limitations of the FEA and other analysis methods used. => Add self-critique.

7 The intended originality is 'In the proposed study a novel methodology to carry out the conceptual design of a pedestrian  and cycle timber bridge was proposed and described, proving its efficacy.' However more work needs to be done to describe (a) what the methodology is, and (b) where it is novel. Also, the claim of ' proving its efficacy ' must be re-considered as there was no actual evidence that this methodology was more effective than another other.

8 The title and conclusions talk about this being a 'timber bridge'. Yet steel is being used, so the title misrepresents the work.

9 The central argument of the paper is that wooden bridges suffer durability issues. I would have expected a critical review of the literature  to determine what factors contributed to durability, and then a more explicit and systematic methodology that showed how to include those in design. As it is the bulk of the paper is taken up with structural calculations, and the durability issues are not prominent. => Reconsider the purpose and structure of the paper.

Author Response

Manuscript ID: applsci-719020,

Title: SOME SHAPE, DURABILITY AND STRUCTURAL STRATEGIES AT THE CONCEPTUAL DESIGN STAGE TO IMPROVE THE SERVICE LIFE OF A TIMBER BRIDGE FOR PEDESTRIANS

Applied Sciences

Special Issue: Assessing and Extending the Service Life of Bridges

ANSWER TO REVIEWERS' COMMENTS

All the new/changed parts in the manuscript are outlined in red. In the following the comments Reviewer 2 are answered. The authors thanks Reviewer 2 for his comments that surely helped to improve the quality of the paper.

Referee: 2
1 English communication is sufficient for comprehension of the authors' intent. => Needs professional proofreading.  

2 Literature review is sparse, and tends to be factual rather than critiquing the body of knowledge. =>  Literature needs work.

Answer: English has been checked. References have been improved.

3 The work is presented as a design report. The methodology is implied rather than explicit. =>  Add a method statement.

5 A detailed analysis of the selected bridge design is presented. For the most part this appears to be design-as-usual. => If there is any novel academic contribution here, it needs to be more explicit.

7 The intended originality is 'In the proposed study a novel methodology to carry out the conceptual design of a pedestrian  and cycle timber bridge was proposed and described, proving its efficacy.' However more work needs to be done to describe (a) what the methodology is, and (b) where it is novel. Also, the claim of ' proving its efficacy ' must be re-considered as there was no actual evidence that this methodology was more effective than another other.

9 The central argument of the paper is that wooden bridges suffer durability issues. I would have expected a critical review of the literature  to determine what factors contributed to durability, and then a more explicit and systematic methodology that showed how to include those in design. As it is the bulk of the paper is taken up with structural calculations, and the durability issues are not prominent. => Reconsider the purpose and structure of the paper.

Answer: The methodology at the basis of the study has been better explained, also by introducing the flow-chart in Fig. 2. It has been also clarified that the novelty of the proposed approach consists in including from the early stage of the conceptual design the analysis of three crucial aspects that can significantly impact the performance of timber bridges during the service life: i) main boundary constraints and load-bearing system; ii) durability; iii) vibration levels. A similar approach cannot be found in literature. So it has been clarified that the methodology focuses on three main topics, and not only on durability issues.

The claim “proving its efficacy” has been deleted.

 4 A number of design solutions were initially considered, and one selected. However we are not informed adequately about how this decision was made. Figures are provided for some wooden models, though these are not particularly informative except in a descriptive way. The overall effect is that the initial stages of concept design appear to have been ad-hoc. => Manuscript needs a better and more rational explanation of the decision-making.

Answer: The decision-making has been better argued in Section 3.

6 There is no critical review of the assumptions or limitations of the FEA and other analysis methods used. => Add self-critique.

Answer: A self-critique has been added in Section 5: “It is worth to note that despite the uncertainties intrinsic in a FE model, the performed analysis, described in the following, well explain the main features of the proposed methodology.”. 

8 The title and conclusions talk about this being a 'timber bridge'. Yet steel is being used, so the title misrepresents the work.

Answer: Even if the final configuration of the bridge under examination is the result of the assemblage of timber and steel elements, wood remains the predominant material and so the diction “timber bridge” has been maintained.

The same approach was adopted in:

Casciati, S.; Faravelli, L.; Bortoluzzi, D. Human induced vibrations in a pedestrian timber bridge. ECCOMAS Thematic Conference - COMPDYN 2013: 4th International Conference on Computational Methods in Structural Dynamics and Earthquake Engineering, Proceedings - An IACM Special Interest Conference 2013, 2609-2618.

Round 2

Reviewer 1 Report

The paper was improved in accordance with recommendation and can be published as it is in the present form.